# Allelochemical Interactions in the Trophic System «*Henosepilachna vigintioctomaculata* Motschulsky—*Solanum tuberosum* Linneus»

**DOI:** 10.3390/insects14050459

**Published:** 2023-05-13

**Authors:** Nathalia Valerievna Matsishina, Marina Vladimirovna Ermak, Irina Vyacheslavovna Kim, Petr Viktorovich Fisenko, Olga Abdulalievna Sobko, Alexey Grigorievich Klykov, Alexey Nikolaevich Emel’yanov

**Affiliations:** FSBSI «FSC of Agricultural Biotechnology of the Far East Named after A.K. Chaiki», Timiryazevsky stl., Volozhenina st., 30 B, 692539 Ussuriysk, Russia

**Keywords:** *Henosepilachna vigintioctomaculata*, dynamics of glycoalkaloids, inhibitor proteinase activity, adrenaline

## Abstract

**Simple Summary:**

The issues of relationships in the system “*Henosepilachna vigintioctomaculata*—potato plant” have not yet been studied. The highest level of stress was demonstrated by larvae feeding on the potato varieties Belmonda, Queen Anne, Lilly, and Dachny, while the lowest stress level was observed when larvae fed on variety Smak. The damage inflicted by potato ladybird beetles on potato leaves led to an increase in the content of glycoalkaloids in some varieties already within 24 h after the transfer of the phytophages. In most cases, the content of glycoalkoloids increased by 20% within five days. Variety Smak did not show a significant increase in the content of alkaloids in response to the damage. It was established that the higher the content of glycoalkaloids and the activity of proteinase inhibitors in the tissues of potato plants, the higher the level of stress in the potato ladybird beetles that feed on them.

**Abstract:**

*Henosepilachna vigintioctomaculata* is an intrinsic element in the agroecosystem of potato fields. The issues of relationships in the system “potato ladybird beetle—potato plant” have not yet been studied. To study the effect of potato varieties on the potato ladybird beetle, only hatched and active larvae with a hatching rate close to 100% were selected from a laboratory colony. Larvae of the first summer generation collected in potato fields were used in our study to determine the level of adrenaline in the bodies of insects, fresh potato leaves were used to study the content of glycoalkaloids, the content and activity of proteinase inhibitors. The larvae that fed on plants of varieties Belmonda, Queen Anne, Lilly, Dachny, Kazachok, Yubilyar, and Avgustin demonstrated the highest level of stress while the stress level in those that fed on variety Smak was the lowest. The damage inflicted by potato ladybird beetles on leaves of some studied potato varieties led to a progressive increase in the content of glycoalkaloids already within 24 h after the phytophages had been transferred. In most cases, the content of glycoalkoloids increased by 20% within five days. Potato ladybird beetles feeding on plants of different potato varieties caused a progressive increase in proteinase inhibitors (% of the control). Plants of variety Smak did not show a significant increase in the content of alkaloids in the herbage in response to the damage. A correlation was established between the mortality rate, the activity of proteinase inhibitors, the dynamics of glycoalkaloids, and the level of adrenaline, which could be formulated as follows: the higher the content of glycoalkaloids and the activity of proteinase inhibitors in the tissues of potato plants, the higher the level of stress in the potato ladybird beetles that feed on them.

## 1. Introduction

It is well known that the potato ladybird beetle *Henosepilachna vigintioctomaculata* Motschulsky, 1857 (Coleoptera:Coccinellidae) feeds on plants from various families [1,2,3,4] but prefers *Solanum tuberosum* L among all others [5]. Currently, there is no consensus on the reasons why phytophagous insects are so selective about host-plants. In the case of the potato ladybird beetle, such studies have never been conducted at all.

However, a number of researchers find the nutritive responses of the Colorado potato beetle similar to those of the potato ladybird beetle [6]. A low consumption rate of the herbage of certain potato varieties by the Colorado potato beetle (*Leptinotarsa decemlineata* Say, 1824) was attributed to their tough and thick leaves with dense indumentum and a high content of ascorbic acid, glutathione, phenolic compounds, demissine, and proteins [7]. Additionally, the mentioned traits of potato varieties reduce the edibility of plants for phytophagous insects [8]. It can be concluded that varieties of agricultural crops with resistance to biotic and abiotic factors determine the character of the specialization of many phytophagous insects drastically changing the insect species composition in ecosystems—attracting insects from wild vegetation to crops or eliminating entire species in the process of replacing native vegetation by cultural one and the formation of agroecosystems [9]. When an insect species switches from wild vegetation to field crops its habitat range and biotic potential usually increase [10]. The 28-spotted potato ladybird beetle is no exception. 

The potato ladybird beetle is a common species of forest fauna. Grasses and trees from the families *Cucurbitáceae, Solanaceae, Fagaceae, Papaveraceae,* and *Amaranthaceae* and the order *Rosáles* were the main host-plants of this pest. The sparse population and the presence of predators prevented the potato ladybird beetle from multiplying exponentially and expanding its habitat range. A direct and indirect influence of people reclaiming the fallow lands covered by broad-leaf and mixed forests led to drastic changes in vegetation and to the formation of potato fields. The process of change in the flora and corresponding fauna of a certain territory, i.e., ecological succession, takes place under the impact of secondary factors. Vast potato fields had a positive effect on the development of the potato ladybird beetle. Leaves of cultivated potato are significantly softer and less acidic than leaves of wild potato species. The potato ladybird beetle quickly switched to the new host-plant turning from a harmless insect into a dangerous pest, whose habitat range increased considerably after that. Consequently, an important role in the spread of the potato ladybird beetle was played by anthropogenic factors, which in combination with the high phenotypic plasticity of the species allowed this pest to populate the whole area of potato cultivation in the south of the Russian Far East. Potato became the main host-plant for this phytophagous insect and a factor in population diversification [5]. The nutrient requirement of a phytophagous insect is the key factor in its interactions with host-plants. This is reflected in the unique nutritional adaptations of insects, which facilitate the effective use of plant material as food. The nutritional adaptation of phytophagous insects is determined by the physiological and biochemical characteristics of both host-plants and phytophages. The resistance of host-plants to phytophages depends primarily on the specific factors of plant immunity, which plays the role of a barrier limiting the suitability of plant species and their organs and tissues for the nutrition of insects and plant-feeding mites [11]. Modern science views the system “phytophaous insect—host-plant” as a result of their coadaptation and coevolution [12]. Searching for suitable plants, feeding on their tissues, and digesting are energy-consuming processes for insects [13]. Therefore, host-plants have a number of specific traits (morphological, physiological, etc.), that prevent phytophagous insects from feeding on their tissues [14]. According to the standard classification, the factors of plant immunity, that produce an adverse effect on the survivability of phytophagous insects, are referred to as antibiosis. The factors of antibiosis include secondary metabolites of plants and their nutritional value. The complex of hydrolyzing enzymes in insect intestines is one of the main targets for plant immunity because the very availability of proteins, sugars, and lipids for phytophages is determined by these enzymes. For this reason, enzymes in the intestines of phytophages play an essential role in their adaptation to the resistance of host-plants to insects. In particular, the composition of enzymes in the intestines of phytophagous insects may change under the influence of proteinase inhibitors synthesized in host-plants as the result of induced resistance to insects. This leads to the avoidance of the influence of these inhibitors. For example, feeding potato leaves treated with jasmonic acid to Colorado potato beetles (i.e., imitating induced resistance to insects), which leads to an increase in the expression of cysteine proteases in the intestines of the beetles. Simultaneously, the synthesis of the inhibitors of aspartic proteases is observed in the treated leaves [15]. The transformation of plant xenobiotic in insects is carried out by the detoxification system [16]. For example, the activity of esterases in the intestine of *Myzus persicae* increases when the insect feeds on tobacco plants with a high content of nicotinic acid compared to pepper plants [17]. The esterase activity also increases when larvae of *Spodoptera litura* feed on plants with a high content of phenolic compounds [18]. An increase in the GST activity (an enzyme of the detoxification system) was detected in the intestines of *Spodoptera frugiperda* and *Trichoplusia ni* when they fed on a substrate with glucosinolates [19]. The glucose oxidase enzyme in the saliva of *Helicoverpa zea* larvae decreases the synthesis of nicotinic acid in *Nicotiana tabacum*, which starts as a response to the damage [20]. As can be noted, the antibiosis is usually activated against larvae and imagines of phytophagous insect species. Consequently, studying the biochemical mechanisms of stress responses in insects and host-plants is necessary to enable their use as regulatory factors in ecosystems. 

Studying how the factors of plant immunity influence phytophagous insects facilitates a better understanding of the limiting effect of fodder species composition on insect populations. The knowledge about the causes and mechanisms of harmfulness formation in the past may serve as a foundation for the prediction of such events in the research on phytophagous populations today. However, data on the interactions in the system “phytophagous insect—host-plant” are not yet sufficient because *Henosepilachna vigintioctomaculata* became a potato pest only relatively recently. To discover the features of such a trophic system, we carried out a number of experiments. The aim of our research was to study the allelochemical interactions in the trophic system “the potato ladybird beetle—potato plant”.

## 2. Materials and Methods

*Maintaining insects in a laboratory colony.* A colony of *Henosepilachna vigintioctomaculata* (Motschulsky) was created in 2019 at the Laboratory of Breeding and Genetic Research on Field Crops (FSBSI “Federal Scientific Center of Agricultural Biotechnology of the Far East named after A.K. Chaiki”). Adult beetles were collected from various locations throughout Primorsky kray (Russia). Insects were collected in their natural habitats for introduction into the culture of the insectarium: on linden *Tilia amurensis*, bird cherry *Padus asiatica*, potato *Solanum tuberosum*, tomato *Solanum lycopersicum*, and eggplant *Solanum melongena*, by selecting ten adult individuals of each sex at different locations in the studied region. Egg masses and larvae of early instars were also collected. The first collection of insects was carried out in 2019, eight laboratory generations were obtained. In 2020 and 2021, the adults collected in nature were introduced into a culture to preserve the polymorphism of the lines. We used standard methods of keeping and breeding insect cultures aimed at optimizing environmental parameters, maintaining density and food supply [9]. When creating the laboratory population, the parameters of minimal mortality, minimal variability of forms, and maximum fecundity were taken into account. To create an ecological optimum, the culture was stabilized, which excluded uncontrollable factors and temporal drift. The dynamics of daily and seasonal temperatures and humidity close to natural were also excluded. Insects were grown at a temperature of 25 ± 1.05 °C and a relative humidity of 85 ± 2.25%, with 16 ± 1.25 h of light per day in fabric insulators. The insulators were placed on racks connected to a timer (time relay). The racks were stocked with grow lights Quantum line ver. 1 (lm281b + pro 3000K + SMD 5050, 660 nm) (Samsung, Seoul, Korea). A constant temperature was maintained by a Rovex RS-07MST1/RS-07MST1 Aux Air split system, Shenzhen, China). Aeration as an element of the microclimate was provided by an AceLine TFSL-6 aerator (Shenzhen, China). The humidity level was controlled using a POLARIS PUH 9105 IQ (Shenzhen, China). In the laboratory, insects were reared on leaves of potato variety Smak, which were grown in soil in a culture room at 25 ± 1.05 °C and a relative humidity of 85 ± 2.25%, at 16 ± 1.25 h of daylight. Potato plants were cultivated in wooden boxes (20 cm in depth) with daily artificial watering (once). Planting density averaged 1.5–2 dm^2^ per plant. The soil was taken from the top layer of coniferous forest soil and carefully sieved to remove plant roots and large soil animals (insects, worms, etc.).

### Experiment Set-Up

*Laboratory experiments.* The following thirteen potato varieties were used in the experiment: Smak, Yubilyar, Kazachok, Sante, Dachny, Avgustin, Yantar, Laperla, Lilly, Queen Anne, Red Lady, Labella, and Belmonda. The wild potato species *Solanum demissum* and *Solanum bulbocastanum* served as the control plants. These wild species originate in ecosystems where the potato ladybird beetle is not present, yet they are recommended for use as the standard in experiments by generally accepted methodology. For the experiment on the effect of potato varieties on the potato ladybird beetle, only hatched and active larvae with a hatching rate close to 100% [21] and without disease symptoms were selected from the laboratory colony. When setting up the experiment, ten larvae of the first generation were placed in each glass container with a volume of 80 mL, containing from one to five leaves of a particular potato variety. Leaves of the potato varieties were collected in experimental field plots. The number of leaves varied depending on the rate of development of the larvae and the food intake rate. A filter paper was also placed in the containers and changed as it was fouled or every two days. The containers were covered with a cotton cloth and placed on shelves in the laboratory. The colonies were observed until the emergence of adult beetles. The following information was recorded: the date and timing of the transition of the larvae from one stage of ontogeny to another, the mortality rate of the larvae, and any observed morphological anomalies and deformations. The time of development and the survival rate of the larvae at each stage was calculated. All experiments were carried out in triplicate.

*Field experiments.* The experiments were carried out under field conditions in a 40 m^2^ experimental field. Potato was planted in ridges of 90 × 30 cm (50 seed tubers per one ridge); the area of each plot was 25 cm^2^. The soil of the field was meadow brown podzolic soil with levelled NPK. Tillage was conducted in late autumn followed by harrowing in early spring, and cultivation before planting. Inter-row cultivation was performed two times during the growing period. Buckwheat was the predecessor in crop rotation. The following thirteen potato varieties were used in the experiment: Smak, Yubilyar, Kazachok, Sante, Dachny, Avgustin, Yantar, Laperla, Lilli, Queen Anne, Red Lady, Labella, and Belmonda. The wild potato species *Solanum demissum* and *Solanum bulbocastanum* served as the control plants. These wild species originate in ecosystems where the potato ladybird beetle is not present, yet they are recommended for use as the standard in experiments by generally accepted methodology. When setting up the experiment, groups of six to eight selected larvae of the first summer generation were transferred onto the upper leaves of individual potato plants. In our study, the larvae of the first summer generation collected in potato fields were used to investigate the content of glycoalkaloids and the activity of proteinase inhibitors in fresh potato leaves, the level of adrenaline in the bodies of insects [22].

To avoid the free movement of the larvae, they were enclosed in calico containers fixed at the base of the leaves. The transferred individuals were under regular observation and removed from plants after they ate approximately 1/3 of the total leaf area [22]. Immediately thereafter, the larvae were eradicated and used for the quantitative determination of adrenaline. The leaves damaged by the larvae were cut with scissors one, three, and five days after the start of the experiment.

The control group, which was selected simultaneously with the damaged leaves, included: (a) undamaged leaves of individually damaged plants; (b) leaves from intact plants. To prevent contamination, calico containers (without potato ladybird beetles) were also fastened around all leaves from the control group and leaves in the experiment. The collected leaves were placed in Petri dishes on wet filter paper for transportation to the laboratory. The collected material was fixed within two hours, and extracts of proteins and glycoalkaloids were isolated from it [23]. To determine the total content of glycoalkaloids in the leaves of potato plants, the accelerated method proposed by Tukalo et al. was employed [24].

The activity of proteinase inhibitors was determined spectrophotometrically by changes in the rate of substrate hydrolysis (∆A/min) with trypsin, a-chymotrypsin, and papain in the presence of potato inhibitor proteins [25]. To quantify the adrenaline level in the potato ladybird beetles, the Ronin method was employed [26], modified to work with larvae and adult insects [27]. Moreover, in the field experiment the date and timing of the transition of the larvae from one stage of ontogeny to another, the mortality rate of the larvae, and any observed morphological anomalies and deformations were recorded. The time of development and the survival rate of the larvae at each stage was calculated. The obtained data were compared to the results of the laboratory experiment, which allowed the performance of correlation analysis.

Statistical processing was carried out using the software PAST v3.17, and such indices as standard deviation, standard error (±SD/±SE), the Shapiro–Wilk test of normality, Spearmen’s R coefficient, and the Tukey’s HSD (honestly significant difference) procedure.

## 3. Results

In our study on the content and dynamics of glycoalkaloids in the intact and damaged leaves of different potato varieties, the physiological state of the ladybird beetles that fed on the potato leaves was evaluated based on the level of adrenaline (Figure 1).

The highest level of stress was demonstrated by the larvae that fed on plants of varieties Belmonda, Queen Anne, Lilly, Dachny, Kazachok, Yubilyar, and Avgustin. The lowest stress level was observed when the larvae fed on variety Smak. In the case of variety Belmonda, the level of adrenaline slightly exceeded that of *Solanum demissum* and amounted to 14.8 mg%. It should be noted that a certain level of stress (determined as an increase in the content of glycoalkaloids), was also detected in potato plants (Figure 2). This is consistent with the postulate of the coevolution in the system “phytophage—potato plant”. In general, in our studies, wild potato species were characterized by a higher content of glycoalkaloids. Cultivated potato varieties differed significantly in the content of glycoalkaloids. Thus, varieties Belmonda, Labella, Queen Anne, Lilly, Kazachok, and Avgustin had a higher content of secondary metabolites compared to the studied wild potato species. However, variety Sante had a considerably lower content; varieties Laperla, Red Lady, and Yantar were similar to Smak, which had the lowest content of glycoalkaloids (120–128 mg/100 g). The damage inflicted on the potato leaves by the ladybird beetle caused in some cases a progressive increase in the content of glycoalkaloids, which was observed already 24 h after the transfer of the phytophage. Thus, variety Belmonda was characterized by an increased content of glycoalkaloids compared to the control group: by 1.3 times higher on the first day, by 1.4 times on the third day, and by 1.5 times on the fifth day. Generally, the content of glycoalkoloids increased by 20% within five days. Unlike most varieties, Smak did not show a significant increase in the content of alkaloids in the herbage in response to the damage.

In undamaged leaves of damaged plants (intact plants), the content of glycoalkaloids did not remain constant, but was rising almost synchronously during the experiment (Figure 3). Thus, the content of glycoalkaloids in the leaves of intact plants of variety Belmonda increased by 1.32 times on the third day. In the case of variety Kazachok, an increase of 1.004 times was detected on the third day, and 1.12 times on the fifth day, which slightly exceeded this parameter value in damaged plants. Variety Smak did not demonstrate any immune responses. Moreover, there was a decrease of 1.18 times in the content of glycoalkaloids on the fifth day.

Mechanical damage substituting for the damage caused by the potato ladybird beetle affected the dynamics of glycoalkaloids in leaves differently depending on the variety (Figure 4).

In addition, the feeding of the potato ladybird beetles on plants of different potato varieties caused a progressive increase in protease inhibitors (% of control). As can be seen from Figure 5, variety Belmonda was characterized by a high activity of inhibitors of trypsin, chymotrypsin, and papain significantly exceeding the standard *S. demissum* in all variants of the experiment. The parameter values of varieties Queen Anne and Dachny were relatively high, while variety Smak demonstrated the absence of any negative impact on the potato ladybird beetle. The content of proteinase inhibitors in plants of variety Smak was lower than in the control plants *S. demissum* (Figure 3). Responding to the damage, variety Smak did not increase the amount of proteinase inhibitors, on the contrary, reduced it. Varieties Kazachok, Labella, Yubilyar, and Lilly were characterized by an increase in the amount of proteinase inhibitors in leaves of undamaged plants (growing beside damaged ones) in comparison to intact leaves of damaged plants (Figure 5). The amount of trypsin inhibitors in the leaves of varieties Dachny, Lilly, and Kazachok damaged by the potato ladybird beetle was 1.5–2.5 times higher than in the leaves of the control plants *S. demissum*. Additionally, the parameter values of variety Kazachok were similar to those of variety Belmonda. The amount of chymotrypsin inhibitors in the leaves of varieties Dachny and Belmonda damaged by the phytophagous insect slightly exceeded the control; variety Kazachok had the parameter values at the same level as the control. Surprisingly, cutting with scissors induced higher activity of papain inhibitors in leaves than the feeding of the phytophagous insect.

We established a correlation between the mortality rate (%), activity of protease inhibitors (%), dynamics of glycoalkaloids (mg/100 g of fresh tissue), and adrenaline level (Table 1, Figure 6). By using the method of pairwise comparisons and constructing correlation trends, a significant linear correlation was revealed between the parameters “adrenaline level—glycoalkaloids dynamics” and “adrenaline level—activity of proteinase inhibitors” (Table 1). This shows that “the higher the content of glycoalkaloids and the activity of proteinase inhibitors in the tissues of potato plants, the higher the level of stress in the potato ladybird beetles that feed on them”. However, correlation trends for other indicators (Table 1) are far from being clear.

Thus, the correlation trend for the pair of indicators “adrenaline level—mortality rate”, although it tended to be linear, had discrepancies in its final part. The pupa weight weakly correlated with the level of adrenaline; the timing of ontogenetic stages did not depend on the level of adrenaline, but it weakly correlated with the dynamics of glycoalkaloids. There was a clear relationship between the indicators “mortality rate—activity of proteinase inhibitors”. The pupa weight did not correlate with either the activity of proteinase inhibitors or the dynamics of glycoalkaloids. These trends indicate the influence of not yet identified factors on the physiological state of potato ladybird beetles. At the same time, the analysis of correlation trends between the indicators “mortality rate—weight of pupae”, “duration of ontogeny—mortality rate”, and “weight of pupae—duration of ontogeny” indicated a weak relationship between fat accumulation and the duration of ontogenetic stages.

Figure 6 shows that potato variety Smak was the most optimal material for the fat accumulation, growth, and development of the potato ladybird beetle at all stages. The mortality rate was the lowest and the fecundity was the highest when the insects fed on this potato variety. The least beneficial varieties for the potato ladybird beetle were Queen Anne, Lilly, Dachny, and Kazachok, which led to the highest mortality rate with shifts in ontogenetic timing. Based on the results of our experiments, variety Belmonda had an advantage over all other potato varieties and demonstrated a complex of immune barriers against the pest. Feeding on this variety led to the highest mortality rate of the beetles with an increase in the duration of ontogenetic stages. A maximum pupa weight was detected when the pupae fed on variety Smak (ā = 54.38 mg) and Yubilyar (ā = 41.5 mg). The pupae had the lowest weight when feeding on variety Belmonda (ā = 12.28 mg). The weight of the pupae was relatively equal when they fed on the rest of the varieties, which was consistent with the mortality rates and the duration of ontogenetic stages. The presented information should be considered a valid and reliable criterion of the high resistance of potato variety Belmonda to the 28-spotted potato ladybird beetle.

## 4. Discussion

Previous research established that glycoalkaloids in potato plants are capable of inhibiting acetylcholinesterase, hindering nerve impulse conduction, and blocking the coordinating activity of the central nervous system, including in insects [28]. Studies on several model objects (suspension cultures of rabbit erythrocytes, beetroot cells, and protoplasts of *Pénicillium notatum* Westling) showed that glycoalkaloids caused cell lysis at certain concentrations, and chakonin in this sense turned out to be almost an order of magnitude more toxic than solanine. The authors of this work suggest that glycoalkaloids change the permeability of cell membranes by incorporating specific steroid compounds into them [29]. As for the potato ladybird beetle and insects in general, their use of some foreign steroids from food to create their own cellular elements (mainly sterols) is the norm due to the inability to synthesize cholesterol on their own [30]. At the same time, the inclusion of other sources of steroid structure in this process, such as steroidal alkaloids and glycoalkaloids, can have fatal consequences for metabolism. This is evidenced by an increase in the level of adrenaline in the potato ladybird beetles that fed on resistant potato samples.

It is known that adrenaline is released into the hemolymph in a state of stress, having a certain physiological effect on the nervous system, which contributes to a change in the period of hyperactivity by a state of prostration. At the same time, adrenaline is involved in the regulation of carbohydrate metabolism in insects, controlling the breakdown of glycogen through the formation of cyclic adenosine-3′,5′-phosphate in the muscles and fat body, as well as the level of free trehalose in the hemolymph [27]. 

The accumulation of glycoalkaloids in leaf cells is induced not only by mechanical damage, but also by biologically active compounds secreted by phytophages when feeding on leaves. Such compounds are common components of insect saliva. It is known that the biosynthesis of sterols and isoprenoids is limited by the amount of the key metabolite of this process—mevalonic acid—and is controlled by 3-hydroxy-3-methylglutaryl-CoA reductase [31]. The activity of this enzyme is regulated at the level of transcription since mRNA formation occurs only in response to cell damage, stress [32], or signaling factors that carry out external regulation of metabolism [33]. In potato, 3-hydroxy-3-methylglutaryl-CoA reductase is encoded by a family of genes whose induction of expression occurs in a strictly differentiated way, depending on what factor it was caused by: a plant pathogen, damage to leaf tissues, or treatment with an elicitor [34].

The components secreted by the salivary glands of insects induce the specific expression of one of the many isoenzymes of 3-hydroxy-3-methylglutaryl-CoA reductase (hmg1 gene), which catalyzes the formation of mevalonic acid. The latter, in turn, being consumed in the isoprenoid pathway, serves as a carbon source for the biosynthesis of the steroid group of glycoalkaloids, which ultimately leads to an increase in their content in cells. At the same time, potato plants are characterized by a high content of inhibitors of proteolytic enzymes in tissues [22]. Serine proteases, especially trypsin and chymotrypsin, perform a digestive function, i.e., break peptide bonds in large proteins with the formation of smaller peptides [35,36]. The inhibition of their activity reduces the weight of larvae and pupae, retards their growth, and prolongs the time of their generation, and these inhibitory effects are more significant at the early stages of development. Thus, in the experiments by Zhao et al. [37], the period of the growth and development of *Plutella xylostella* that ingested various concentrations of protease inhibitors differed significantly compared to the control group. The results of this study agreed with the opinion of Ortego et.al. [38] that various protease inhibitors have different levels of inhibition on insect growth, and the periods of growth and development depend on the concentration of the inhibitor. However, insects have a certain ability to adapt. Thus, when *Prodenia litura Fabricius (Lepidoptera Noctuidae)* was fed for several generations with food containing a large amount of trypsin inhibitors, the inhibitory effect on larva weight and pupa weight decreased [39]. At the same time, Wang and Qin [40], Upadhyay and Chandrashekar [41] showed that the higher the inhibitor concentration, the more obvious the inhibitory effect: if insect larvae ingested low concentrations of protease inhibitors for a long period chymotrypsin activity increased, the activity of high alkaline trypsin decreased, while the activity of the total protease remained unchanged. In addition, protease inhibitors upset the balance between proteases, causing a disorder in the digestive system, which affected the growth and reproduction of insects [42]. Oppert [43] found that for mite pests—*Tribolium castaneum* Herbst (Coleoptera: Tenebrionidae)—control was more effective when serine and cysteine protease inhibitors were used simultaneously. In our studies, we also observed that varieties with different activity of protease inhibitors had different suppressive effects on protease activity and, consequently, extended the growth and development periods of the potato ladybird beetle [44].

Our research results are consistent with the data published by other scientists and support the existence of intensive interactions in the system “phytophagous insect—host plant”. For example, plants synthesize specific metabolites, which are able to affect insects, as a response to the damage caused by phytophages. In our case, this effect was observed through a change in the dynamics of glycoalkaloid and adrenaline accumulation. Therefore, the immune responses of plants are not constant and depend on the genotype of a particular variety. Cultivated potato as a food resource for phytophagous insects is highly diverse in immune responses, which in their turn are determined by the breeding process. 

Based on our data, the immune responses of plants could be divided in three quantitative categories: (1) the absence of metabolite accumulation or very low accumulation producing no effect on phytophages (Smak, Yantar, Red Lady, Laperla); (2) moderate accumulation of metabolites significantly effecting the ontogeny of phytophages (Sante, Yubilar, Kazachok); and (3) high accumulation of metabolites resulting in a strong adverse effect on the ontogeny, survivability, and stress level of phytophagous insects (Labella, Dachny, Queen Anne, Avgustin, Lilly), and even in absolute antibiotic effect (Belmonda). 

Thus, active compounds in food are the basis of the antibiotic effect on phytophagous insects and can cause functional changes in certain systems of their organisms and overall stress. Our experimental data support this hypothesis demonstrating different adrenaline levels in the potato ladybird beetle depending on the immune responses of the studied potato varieties. Plant immune barriers lead to deficiencies in nutrient intake and the development of heterochronies in phytophages during the ontogeny. The next possible step could be the use of protease inhibitors as new biological pesticides to control the potato ladybird beetle or the transfer of the protease inhibitor gene(s) into the host-plant to reduce the nutrient intake by the phytophage, thus suppressing its development and minimizing damage to host plants.

## 5. Conclusions

The presented data revealed a strong connection between the development of individual beetles, the dynamics of potato ladybird beetle populations and potato plants. A potato plant as the base of the trophic pyramid affects the next level, i.e., its phytophage, and determines its functional state impacting the adaptation of the insect to unfavorable environmental conditions. The quality of nutrition, which depends on the content of primary and secondary metabolites, is key to the influence produced by plants on insects. Our data demonstrated the induced resistance of potato plants to the damage caused by the potato ladybird beetle. The magnitude of such a response directly correlates with the degree of damage. The research results showed that the inhibition of digestive enzymes in insect intestines could be one of the possible mechanisms of a direct influence on the induced resistance of potato plants to phytophages. In addition, the total content and accumulation dynamics of glycoalkaloids in herbage in response to the damage caused by the potato ladybird beetle as well as glycoalkaloid composition are determining factors of the immune response in potato, directly affecting the metabolism of the phytophage and resulting in its decreased survivability. 

## Figures and Tables

**Figure 1 insects-14-00459-f001:**
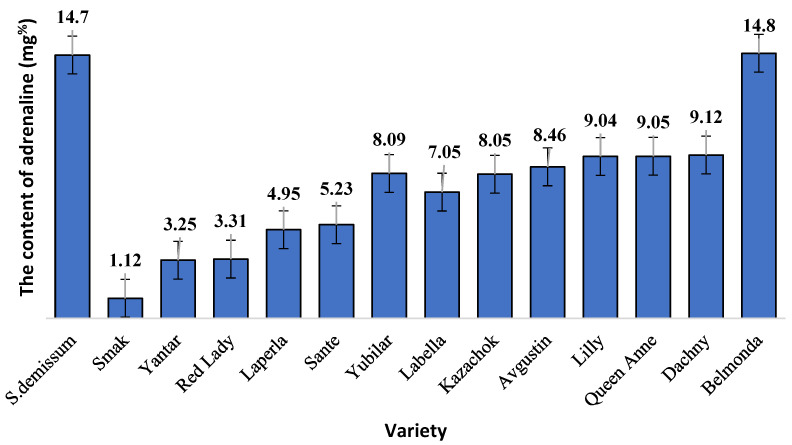
The adrenaline level (mg%) in the bodies of larvae of the potato ladybird beetle feeding on different potato varieties; ±SD 95% confidence interval.

**Figure 2 insects-14-00459-f002:**
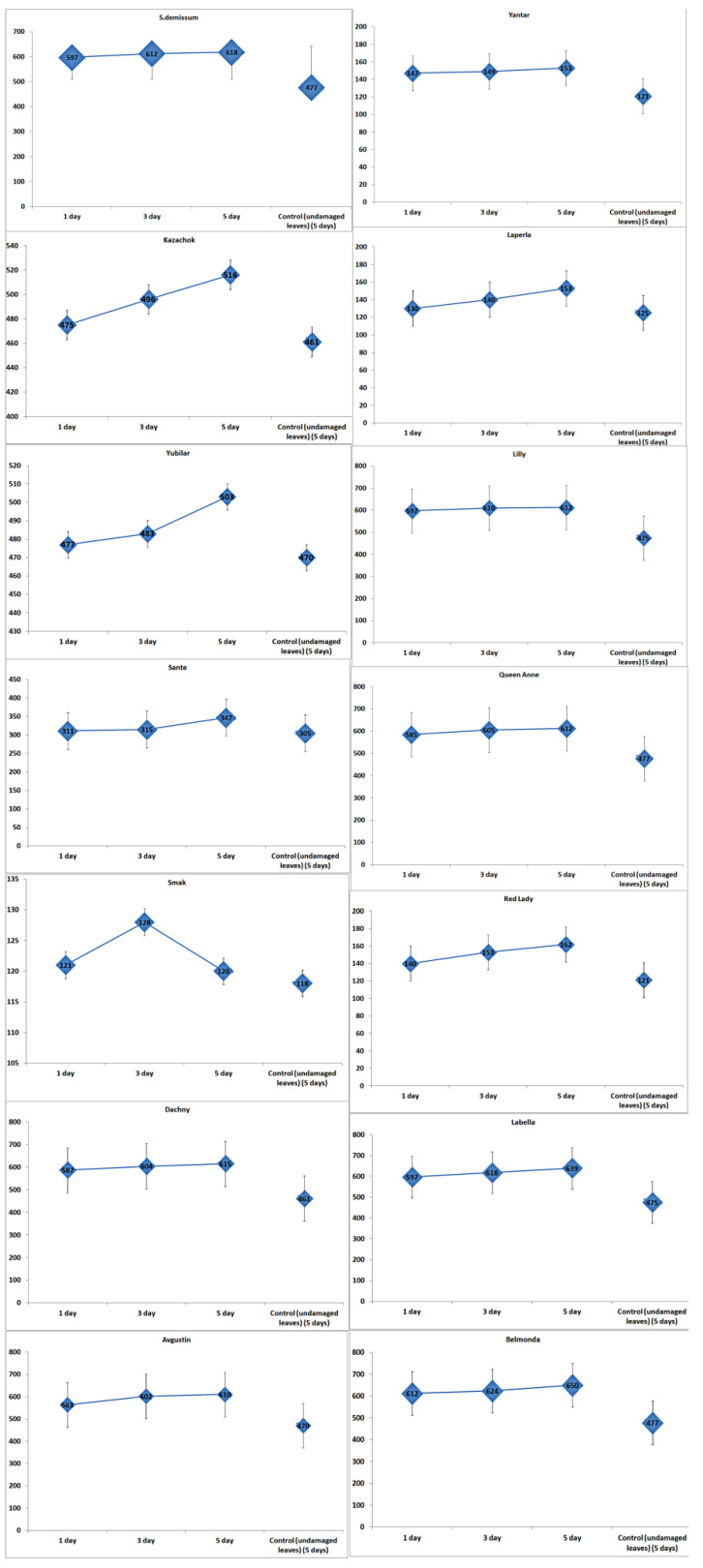
The content and dynamics of glycoalkaloids (mg/100 g of fresh tissue) in the leaves of potato plants damaged by the potato ladybird beetle. Note: based on the presented information, the significance level was α = 0.05 and the degrees of freedom were df_1_ = 3 and df_2_ = 3. The rejection region for the F-test was R = {F:F > 2.769}. Judging from the information about the sample, we concluded that F = 0.626 ≤ F = 2.769. Therefore, there is not enough evidence to claim that not all means were equal at a significance level of α = 0.05. Q value indicated a significant result.

**Figure 3 insects-14-00459-f003:**
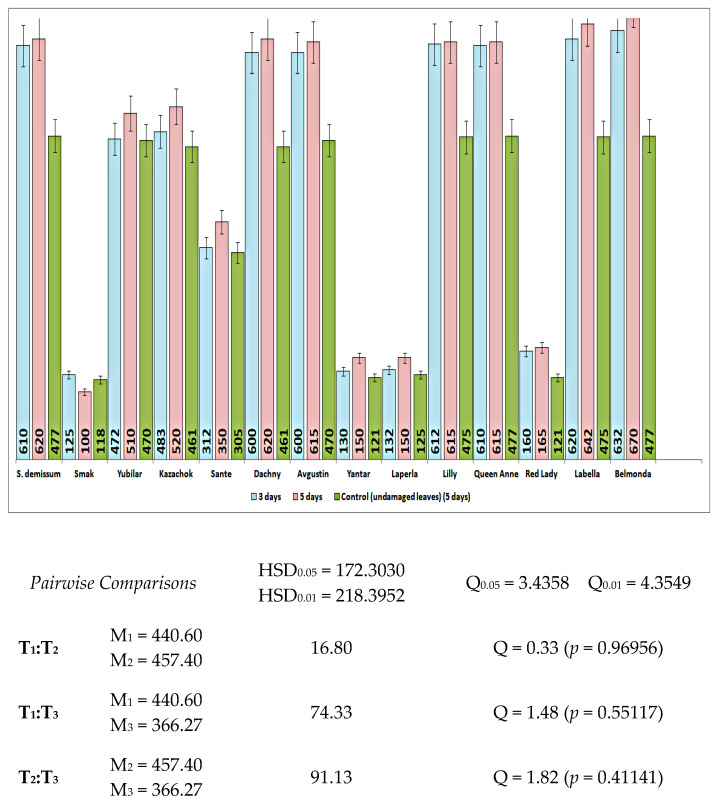
The content and dynamics of glycoalkaloids (mg/100 g of fresh tissue) in the leaves of intact potato plants; the f-ratio value is 0.93527. The *p*-value is 0.4005.

**Figure 4 insects-14-00459-f004:**
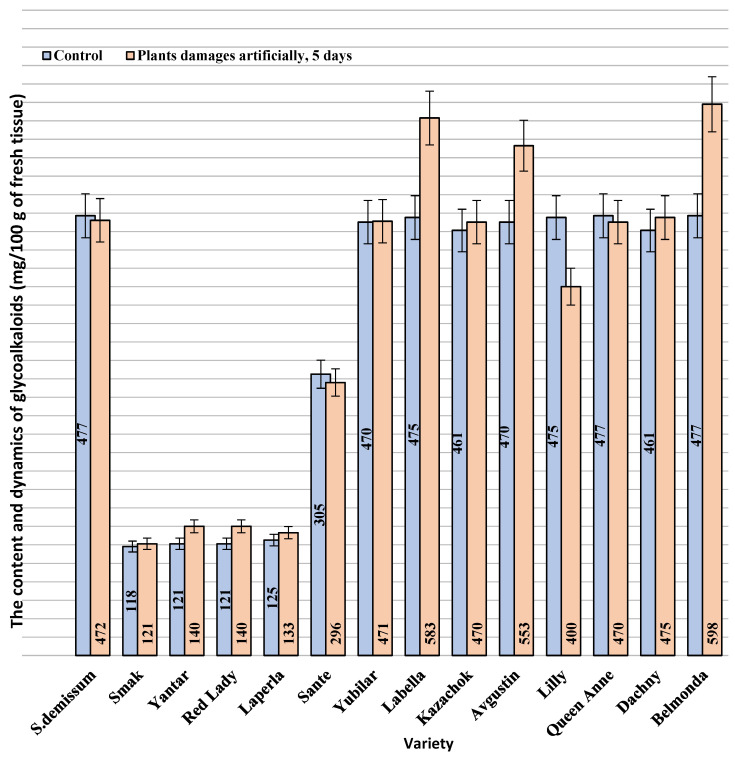
The content and dynamics of glycoalkaloids (mg/100 g of fresh tissue) in the leaves of artificially damaged potato plants. The f-ratio value is 0.07557; the *p*-value is 0.785408.

**Figure 5 insects-14-00459-f005:**
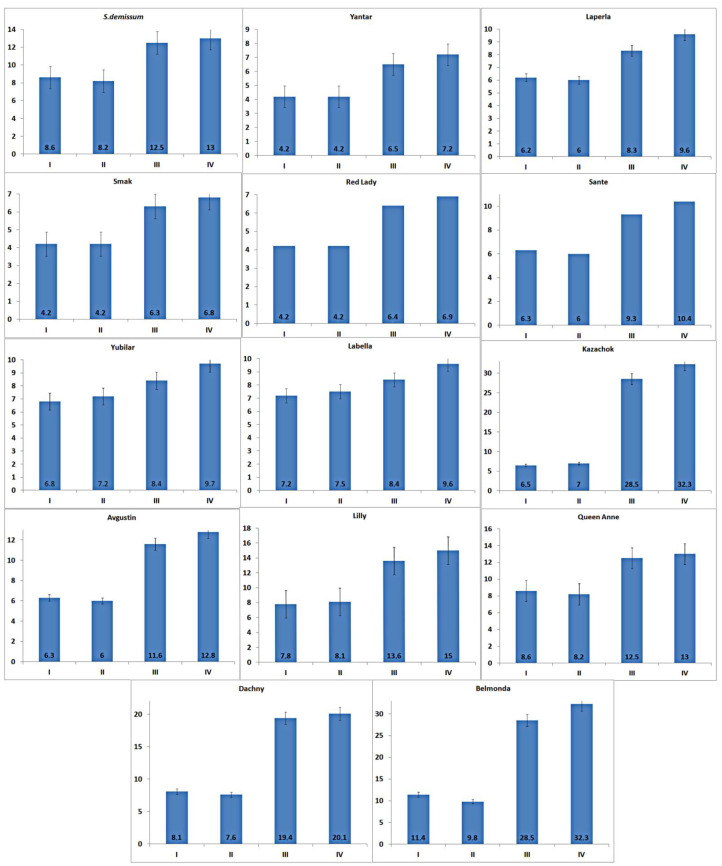
Activity of proteinase inhibitors (%) in the leaves of damaged and intact potato plants. Note: I—intact leaves of damaged plants; II—leaves of intact plants; III—leaves damaged with scissors; IV—leaves damaged by potato ladybird beetles; ±SD 95% confidence interval.

**Figure 6 insects-14-00459-f006:**
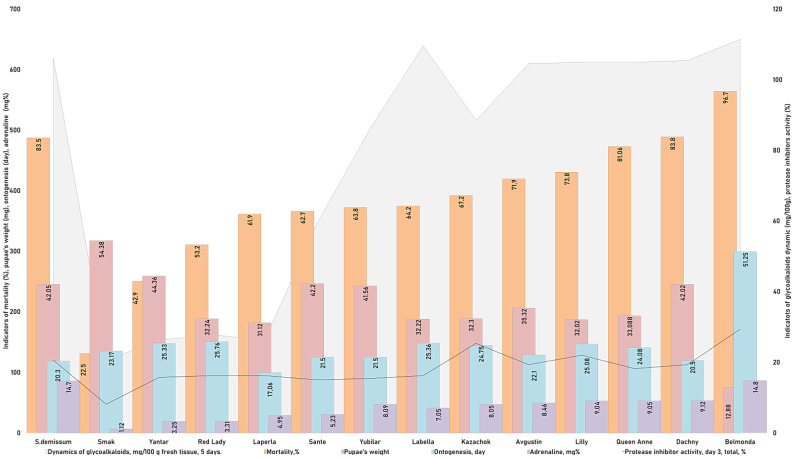
Interactions in the system “potato ladybird beetle—potato plant”.

**Table 1 insects-14-00459-t001:** The correlation analysis of the studied parameters by Spearman’s method (R), *p* ≤ 0.01.

R/R^2^	The Level of Adrenaline (mg%)	The Content and Dynamics of Glycoalkaloids (mg/100 g of Fresh Tissue)	Activity of Proteinase Inhibitors (%)	Mortality (%)	Pupa Weight, mg	Ontogeny, day
Mortality	0.9345/0.8733	0.8143/0.6631	0.6536/0.4272	-	−0.7556/0.5709	0.4464/0.1993
Pupa weight	−0.7619/0.5805	−0.4913/0.2414	−0.7404/0.5482	−0.7556/0.5709	-	−0.7144/0.5104
Ontogeny, day	0.6134/0.3763	0.2038/0.0415	0.5976/0.3571	0.4417/0.1951	−0.7146/0.5107	-
The level of adrenaline (mg%)	-	0.86/0.7396	0.3958/0.1567	-	-	-

## Data Availability

No new data were created or analyzed in this study. Data sharing is not applicable to this article.

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
