# Peer review of "Allelochemical Interactions in the Trophic System «Henosepilachna vigintioctomaculata Motschulsky—Solanum tuberosum Linneus»"

_insects, 2023, doi:10.3390/insects14050459_

Round 1
Reviewer 1 Report
The manuscript contains interesting data, unfortunately it is poorly written. The introduction takes up a lot of space in the account of how this species spread, which does not seem particularly important in the context of this study. Instead, the introduction provides little focus on the research objective and how this fits into current knowledge. I recommend doing a check of the most recent references and leaving citations of Russian-language articles only when strictly necessary.
The materials and methods greatly detail some aspects while others are unclear. For example, I cannot tell exactly which experiments were done in the laboratory and which in the field. In the field, the different potato varieties were grown under what conditions? Was an experimental field set up? Were the insects used in the field those reared in the laboratory? Were tests on developmental duration and survival provided by feeding the insects with leaves of the plants grown in the field or of plants kept in the laboratory? Because the materials and methods are unclear, it is difficult to judge whether the experimental design is appropriate
In the materials and methods, no mention is made of the tests used in the statistical analysis. Only the software used is indicated. Nor are the results of the statistical tests clearly shown in the results section. I have difficulty in understanding the graphs. Graphics are complicated. In addition, graphics lack axis titles and units of measurement.
The discussion section suffers from the lack of clarity of the objectives of the study.
I believe that the manuscript can be considered for publication only after it has been rewritten
Author Response
Dear Reviewer! Colleague!
We wish you a good day.
We sincerely thank you for your work and for reading our manuscript and for your comments. This is very valuable to us. All your comments are taken into account.
- The manuscript contains interesting data, unfortunately it is poorly written. The introduction takes up a lot of space in the account of how this species spread, which does not seem particularly important in the context of this study. Instead, the introduction provides little focus on the research objective and how this fits into current knowledge.
We did not shorten the introduction, because the history of becoming a pest in the 29 spotted ladybird is important for understanding the processes occurring in the population.
The reasons of mass reproduction of phytophagous insects or their transition from the category of episodic pests to chronic ones have long attracted the attention of many researchers. Until recently, the search for these causes was focused on the analysis of the role of exogenous factors. Insufficient attention was and is paid to the role of fodder plants not only as an external factor, but also as the most important factor determining the basis of the vital activity of the first order consumers. The formation of all categories of food specialization of consumers (gostal, ontogenetic and topical) as well as the specificity of autoecological and population dynamic characteristics of consumers, including ecological and physiological reactivity and the course of processes of diversification and divergence of their populations are connected with the functioning of immunogenetic system.
In our introduction we wanted to show that the potato ladybird is a relatively recent (in evolutionary perspective) pest, and the above mentioned processes are only proceeding in its population.
- I recommend doing a check of the most recent references and leaving citations of Russian-language articles only when strictly necessary.
In this case, we use the Russian-language literature only when absolutely necessary, as references to the methods used to set up the experiment. We apologize for this, but in this case we consider it difficult to substitute literary sources. This could lead to falsification of the data.
- The materials and methods greatly detail some aspects while others are unclear. For example, I cannot tell exactly which experiments were done in the laboratory and which in the field. In the field, the different potato varieties were grown under what conditions? Was an experimental field set up? Were the insects used in the field those reared in the laboratory?
Fixed. The experiments were carried out under field conditions in a 40 m2 experimental field. Potato was planted in ridges of 90Ñ…30 cm (50 seed tubers per one ridge); the area of each plot was 25 cm2. The soil of the field was meadow brown podzolic soil with levelled NPK. Tillage was conducted in late autumn followed by harrowing in early spring, and cultivation before planting. Inter-row cultivation was performed two times during the growing period. Buckwheat was the predecessor in the crop rotation. The following thirteen potato varieties were used in the experiment: Smak, Yubilyar, Kazachok, Sante, Dachnyi, Avgustin, Yantar', Laperla, Lilli, Queen Anna, Red Lady, Labella, and Belmonda. When setting up the experiment, groups of 6-8 selected larvae of the first summer generation were transferred onto upper leaves of individual potato plants. In our study, larvae of the first summer generation collected in potato fields were used to investigate the content of glycoalkaloids and the activity of proteinase inhibitors in fresh potato leaves, the content of adrenaline in the bodies of insects.
- Were tests on developmental duration and survival provided by feeding the insects with leaves of the plants grown in the field or of plants kept in the laboratory?
For the experiment on the effect of potato varieties on the potato ladybird beetle, only hatched and active larvae were selected from the laboratory colony with a hatching rate close to 100% and without disease symptoms. When setting up the experiment, ten larvae of the first generation were placed in each glass container with a volume of 80 ml, containing from one to five leaves of a particular potato variety. Leaves of the potato varieties were collected in experimental field plots. The number of leaves varied depending on the rate of development of the larvae and the food intake rate. A filter paper was also placed in the containers and replaced as it was fouled or every 2 days. The containers were covered with a cotton cloth and placed on shelves in the laboratory. The colonies were observed until the emergence of adult beetles, and the date and timing of the transition of the larvae from one stage of ontogeny to another, the mortality rate of the larvae, any observed morphological anomalies and deformations were recorded. The time of development and the survival rate of the larvae at each stage was calculated. All experiments were carried out in triplicate.
- In the materials and methods, no mention is made of the tests used in the statistical analysis. Only the software used is indicated. Nor are the results of the statistical tests clearly shown in the results section. I have difficulty in understanding the graphs. Graphics are complicated. In addition, graphics lack axis titles and units of measurement.
Thank you for that observation. Corrected.
- The discussion section suffers from the lack of clarity of the objectives of the study.
Please explain this postulate. I beg your pardon. However, in the "Discussion" section, we strictly discuss the results obtained - the dependence of changes in the level of adrenaline in the phytophage body on feeding on potato varieties; and the dependence of the accumulation of glycoalkaloids and changes in the level of protease inhibitors in the plant on phytophage feeding.
Once again, I apologize.
- I believe that the manuscript can be considered for publication only after it has been rewritten
All in all, we are indeed very grateful for your comments, and we hope for a positive decision. However, your esteemed colleague, the other reviewer sees no need for a complete reworking of the manuscript. Since we must consider his opinion as well, we have decided to leave the content and structure unchanged. However, as you can see, we have addressed most of your comments.
We apologize for the inconvenience.

Reviewer 2 Report
This paper offers insight into interactions between the potato ladybird beetle and its host Solanum tuberosum L. These interactions have not yet been studied and include potato cultivars with different levels of resistance to this pest. Generally, the paper is well written, the presentation of the Results is clear, and the Discussion gives a consistent interpretation of the main results.
The eventual weakness of the manuscript is the absence of insect digestive proteinase analysis since the authors showed protease inhibitors activity in damaged potato leaves. Although, generally, insects have to pay a metabolic price to overcome the negative effect of the plant defense by a reduction in fitness - as a result of co-evolution insect digestion can be relatively unaffected by host plant protease inhibitors, despite its increase by insect feeding. Thus, in their paper authors should not mix the terms protease inhibitors activity with the activity of proteinases.
Other issues that need the author's attention:
Lines 9 and 20: "forage plant" - I'm not sure that this term is adequate for potato. A crop plant, or potato plant, for instance, would be more suitable.
Lines 24-25: Please be more precise. Glycoalkaloids and protease inhibitors (PIs) are parameters for potato, while adrenaline and mortality rate for insects.
Line 30: What is with PIs increase?
Line 32, 34 and Keywords: It is actually PIs activity, rather than the activity of proteinases.
Line 160-161: Please rephrase, for instance … glycoalkaloids and the activity of protease inhibitors in fresh potato leaves, and the content of adrenaline in the bodies of insects.
Line 169-170: Please make clearly distinguish between these control groups, i.e. undamaged leaves from damaged plants and leaves from intact plants.
Line 180: "different potato tissues" - from different groups of potato leaves?
Line 218: "intact plants" - intact leaves?
Lines 234-239: Please describe these PIs increase in some more detail, as it was done for the content of glycoalkaloids.
Lines 253 and 257: PI activity.
Figure 7:
- I think that title: Interactions in the system "potato ladybird beetle - potato plant" would be more adequate for the presented parameters.
- Please add units for the Pupae's weight
Figures 13, 17. and 19: Protease inhibitors activity
Lines 271-272: Protease inhibitors activity
Lines 351, 352, and 360: "forage plant" - Crop plant or potato plant, for instance.
Author Response
Dear Reviewer! Colleague!
We wish you a good day.
We sincerely thank you for your work and for reading our manuscript and for your comments. This is very valuable to us. All your comments are taken into account. We have added the conditions of the experiments, signed the axes of the graphs where necessary; in addition, we have expanded the description for protease inhibitors in the results section.
All in all, we are indeed very grateful for your comments, and we hope for a positive decision.

Round 2
Reviewer 1 Report
I am sorry that the authors resented my comments. I did not mean to be disagreeable. However, the introduction continues to seem weak to me. Does this study intend to show that there was/isn't an adaptation of the beetle to potato or that plant resistance can be an effective control tool? As an evolutionary study it is very weak. There is no population comparison or plasticity data. In contrast, as a study of plant resistance and physiological effect on the beetle, it is rich in data. The discussion, correctly, is focused on aspects of the plant-phytophage relationship. The study highlights a significant varietal difference in the response to the phytophage. The identification of resistant cultivars is very important in the control perspective. However, reading the introduction, one would expect the data to be discussed from an ecological or evolutionary perspective but, in the end, this does not seem to be the focus of this research. The history of becoming a pest in the 29 spotted ladybird is important for understanding the processes occurring in the population but how do these processes relate to this study?
Otherwise, the manuscript has improved.
Author Response
Good afternoon, dear colleague.
We thank you sincerely for your comments. They are indeed very important to us.
We have rewritten the introduction according to your recommendations.
Thank you for your help and we apologise for the inconvenience.